# Social Monetary Valuation for Protecting Forests and Protected Wild Animals in North Sulawesi, Indonesia

**Jerry Mauri** [1]ⓘ**, Yingli Huang** [1,*]ⓘ **and Jun Harbi** [1,2]ⓘ

[1] Forestry Economics and Management College, Northeast Forestry University, No. 26 Hexing Road, Harbin 150040, China; jmauri.01@outlook.com (J.M.); junharbi@gmail.com (J.H.)
[2] Forestry Program Study, Faculty of Agriculture, Muhammadiyah University of Palembang, Jl. Jend. A. Yani, Dua, 13 Ulu, Palembang 30263, Indonesia
[*] Correspondence: ylhuangnefu@163.com; Tel.: +86-13359995881

**Abstract:** Many protected wild animal species are threatened with extinction because of degraded forest habitats. We conducted a monetary assessment of social responses to this issue in North Sulawesi, Indonesia. Respondents were asked to determine a monetary value for forest services, and we measured willingness to pay (WTP) using the collection of compensation funds every year for five years. A descriptive statistical model, a correlation analysis, and a double-bounded dichotomous choice (DBDC) model were used in this study. There were 92.1% of respondents who claimed to be aware of the environment, as well as 96% of protected animals, and 89.8% who agreed with the compensation fund. There is a need for current socialization of the environmental situation, and it is necessary to consider education and income factors for real actions in the future. The estimated monetary value probability for WTP was determined using accepting the median estimate of IDR 264,820 (USD 18.26). These results estimated the value of annual forest protection over five years.

**Keywords:** ecosystem service; forest protection; monetary valuation; social assessment; protected wild animals

## 1. Introduction

The ecosystem is an inseparable part of our lives, so protected forests and wildlife play an important role in the social and economic well-being of communities. Forests provide various ecosystem services, such as water purification, climate regulation, and conservation of biodiversity. Protected wildlife is a source of cultural and spiritual meaning [1], as well as a source of income from tourism [2].

However, protected forests and wildlife face several threats, such as habitat loss due to illegal hunting [3], disasters like floods [4], fires [5], and droughts [6], and even land conversion projects that are not suitable for forest characteristics [7]. To classify protected wildlife, the Government of Indonesia has regulated the characteristics of protected animals based on several conditions, such as limited distribution of endemic wildlife in their habitat, declining populations, habitat destruction, exploitation, and irresponsible use [8].

Global forest degradation and the threat to wildlife have become major environmental issues. From 2000 to 2012, 2.3 million square kilometers of forest were cleared, contributing to the biodiversity crisis, with over one million species facing extinction [9]. Deforestation has also caused the loss of habitats for many species, contributing to the current global biodiversity crisis where over one million species are at risk of extinction [10]. Indonesia has lost over 6 million hectares of forest cover from 2000 to 2012, causing significant environmental problems such as soil erosion, floods, and loss of wildlife habitats. Deforestation threatens the survival of endangered species like orangutans, tigers, and elephants and impacts the livelihoods of local communities that depend on forests for food, medicine, and income [11]. In North Sulawesi, many protected species, such as the Sulawesi Crested

Macaque, Sulawesi Bear Cuscus, Spectral Tarsier, Anoa, Maleo Birds, and Babirusa, are at risk of extinction due to habitat loss [12].

A Science study emphasizes the urgent need for transformative change due to human activities decreasing life on Earth, with land use change, climate change, and resource exploitation being major causes [13]. Additionally, an Ecological Economics study underscores the importance of sustainable economic models like circular economics to prevent species extinction. These studies suggest a crucial role for sustainable approaches and collective efforts in future environmental protection [14].

A reference for this study analyzed orangutan population data and conservation investments in three locations over a 20-year period, funded consistently by the government, and found that conservation efforts successfully increased the orangutan population. Factors such as the presence of primary forests and distance from human settlements contribute to the survival of orangutans. However, further conservation efforts are needed for the long-term survival of orangutans [15]. This is necessary because, despite government attention to forest conservation, heavy reliance on government funding is a common issue. Government funding is vulnerable to sudden changes in funding priorities and budget cuts and exploring alternative financing strategies [16,17].

Apart from government funding, there is also funding that relies on non-government organizations for community well-being activities with building long-term market-based mechanisms for ecosystem and environmental conservation [18,19]. Community involvement in preserving protected wildlife is driven by sensitivity, awareness, and concern for extinction [20,21]. Protecting the "home" of these protected wild animals, especially forests, is one way to ensure the survival of endangered species [22]. Ecosystem services and benefits are inherently closely related. For example, primary and secondary forests provide benefits such as food availability and clean air via air-filtering ecosystem services provided by trees and other plants. This environmental availability supports animal conservation. Conventionally, the measurement of non-SNA (System of National Accounts) benefits for ecosystem accounting purposes is limited to ecosystem service flows with identifiable links to human well-being [23].

Therefore, this study aims to address this knowledge gap by exploring the monetary value communities are willing to contribute towards conservation initiatives in the region. The findings can inform evidence-based policymaking to develop conservation strategies that are ecologically sound, socially accepted, and economically viable. This will ultimately enhance the success and sustainability of conservation in North Sulawesi.

This article discusses a research project that aims to investigate the willingness of people in North Sulawesi to pay or donate to protect forests and protected wildlife. The study has been conducted using a sample of native or locally-born individuals in North Sulawesi and has been run for approximately 6 months during the COVID-19 pandemic.

This study uses the Willingness to Pay (WTP) method to estimate the financial value of threatened forests in North Sulawesi [24,25], which is a survey-based method. The Contingent Valuation Method (CVM) is then used to determine how much people are willing to pay to restore degraded ecosystem conditions. The aim is to evaluate the intangible costs of economic conditions to avoid problems and address disturbances that arise during monetary activities [26].

The study also explores the concept of contribution, which refers to willingness to pay and involves examining respondents' opinions on what they know about the subject of investigation. Finally, the results of this research are in line with what has been performed previously, namely that the literature shows a positive relationship between certainty scores and respondents' prior knowledge about specific value items or attitudes towards hypothetical markets [27,28]. Examples include financial contributions made by individuals and businesses to environmental non-profits [20,29].

Our previous study entitled "Monetary Valuation of Protected Wild Animal Species as a Contingent Assessment in North Sulawesi, Indonesia" [30] with this study used the same questionnaire data set, but we divided the questionnaire into two parts with different

questions for each paper. We conducted different data analyses for the first and second papers, and these two papers are a series of research units.

## 2. Study Area and Methodology

In assessing national forest accounts, the geographical scope of the account must be clearly stated in the boundaries of the land area and the boundaries of the waters entering the territorial boundaries. However, there may be good reasons to expand coverage to include areas under national administrative control [23]. This research was conducted in North Sulawesi (Sulut), which is the northernmost province of the Republic of Indonesia that borders the sea with the Philippines. The island of Sulawesi is an elongated peninsula with a geographical location of 0° N–3° N and 123° E–126° E and is one of the areas of Indonesia that is located north of the equator (Figure 1). The population of North Sulawesi in 2020 was 2,621,923 people, and it has an area of 15.069 km$^2$ [31,32]. The island of Sulawesi lies between the Wallacea and Webber lines that separate the Indo-Malay biogeographical area in the west and Austrasia in the east. This situation makes Sulawesi blessed with various unique types of flora and fauna that cannot be found in any other area [33].

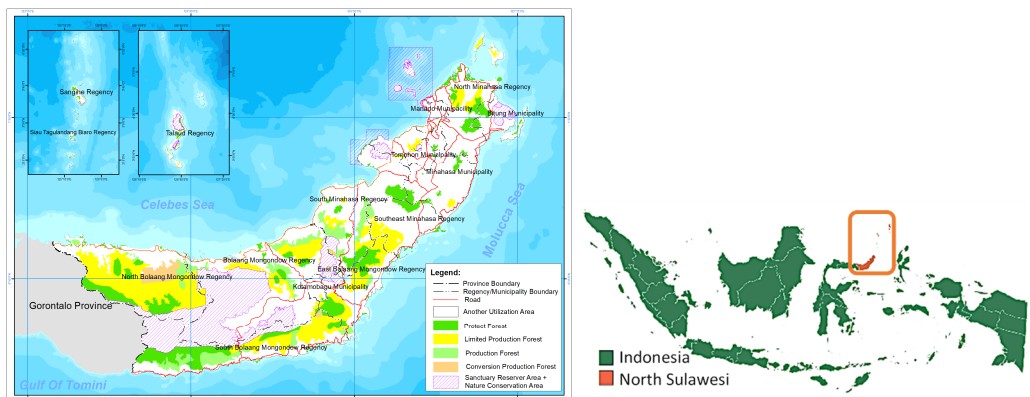

**Figure 1.** Forest cover of North Sulawesi Province, the site for research on social, and monetary valuation [25].

The survey was conducted in North Sulawesi province in 2020. The sample obtained from positive verification was 428; this included distributed online and offline questionnaires. North Sulawesi province consists of fifteen regions, namely four municipalities and eleven regencies. The respondent data obtained offline and online by the city and municipality are shown in Table A1 in Appendix A [30].

Access difficulties caused researchers only to access 11 of the 15 representative region samples in this study. However, one respondent from East Bolaang Mongondow Regency (Kab. Bolmong timur) participated via an online questionnaire, changing the sample to include 12 regions. Research subjects included respondents' socioeconomic problems related to forest protection and the conservation of protected areas [33]. The questionnaires were distributed using the purposive sample method [34].

In the study stage, the selection of the study area was the first task to complete. The selection of area was carried out on the grounds that the characteristics of flora and fauna fell between two biogeographical areas, with most of the forest not yet part of the industrial plantation area. The people were friendly, and the majority were cooperative in expressing their opinions. Other procedures of this research included the following steps (Appendix C.

The dichotomous choice contingency valuation method (DCCVM) has become the most popular technique among practitioners of the contingency valuation method [35]. The DCCVM was used to obtain public WTP for a fair price estimate [36] to contribute to the monetary assessment of forest sustainability.

In general, the CVM, including the DCCVM [37,38], is subject to various biases, such as subjectivity in the initial bid design, the influence of a respondent's psychological

and external conditions, hypothetical directions that lead to a higher WTP, operational respondents who do not understand what is being assessed, and foreign assessment tool. Nevertheless, the survey approach itself is not difficult to perceive as part of a market transaction that is comparable to what consumers face [39].

The assessment of socioeconomic characteristics was conducted by calculating the mean and median of the respondents. The evaluation is expressed based on the categories of each variable, where the range of the mean and median serves as the benchmark for the expected results.

When this elicitation method is used, respondents are only asked to answer YES or NO when asked whether they are willing to pay (offer) a certain amount for public goods [39]. A single-bound model only consists of one question, while in a double-bound model, the first question is followed by other questions with not too many numbers. This procedure is undoubtedly more straightforward for respondents than other methods that require a long adjustment process. It is like a bidding game, as is the case in the open elicitation techniques for the appropriate assessment of individual maintenance costs based on thoughtful analysis. The price paid for this is the limited information from the DCCVM data: the only information available to a researcher after the questionnaire is the value interval containing the actual WTP. We then used this information in the RStudio v.1.4.1106 script using a double-bounded dichotomous choice (DBDC) model. DBDC has two methods available to calculate confidence intervals for WTP forecasts, namely the Krinsky–Robb and Bootstrap methods. Conceptually, the simulation steps of the Krinsky–Robb method rely on the distribution of theoretical simulations. In contrast, the Bootstrap method relies on the distribution of sample data [40]. The difference between these two methods only appears in the lower and upper bound results but does not affect the mean and median results. The Krinsky–Robb method used a truncation mean technique to remove extreme values, while the Bootstrap method used a truncation mean and resampling to obtain a sampling distribution of the data. Both methods had the same estimated means, truncated means, and medians [41]. Differences in value in the LB and UB occurred in the means, truncated means, and medians due to the different sampling techniques used by each method. This difference in sampling technique resulted in different LB and UB values [42].

In estimating the DBDC model, it was necessary to conduct some further data management. The data frame contained the number "1" if the respondent answered "yes" and the number "0" if the respondent answered "no". In WTP, the second question depends heavily on the first question. For the first answer, if the respondent answered "yes", then the number of offers in the second question increased; on the contrary, in the first question, if the respondent answered "no", then the number of offers in the second question decreased [43].

The econometric model we adopted was proposed by Cameron and James (1987), with this model allowing us to calculate the mean or median values of WTP easily [44]. Moreover, the standard error of the coefficients was directly calculated in an analytical formula with the aggregate of the existing confidence intervals. Thus, it included the WTP estimate in the contingent assessment study. The econometric model of the logistic assumptions used for the linear functional form of the median WTP was as follows:

$$Y_i = x_i' \beta + \varepsilon_i \tag{1}$$

In the model, "$Y_i$" is considered a latent continuous, censored variable, where "$Y_i$" is the individual who is willing to pay. This assumption is closely related to the socioeconomic characteristics of the individual expressed in the vector "$x_i'$". The vector $x_i'$ contains the predictor variables that are common to both outcomes, with the coefficient estimator $\beta$ [45].

"$\varepsilon_i$" is the error term distribution in the c.d.f (cumulative distribution function), and F($\varepsilon_i$) is a zero mean with variance equal to $v^2$. The observation variable focused on the answer of "yes" or "no" from each individual who was asked to pay a certain amount of rupiah (IDR).

When observing two dichotomous variables, a double-bound model is used: namely, there is a follow-up action in the form of a second question after the answer to the first question is given [46]. The log function for the double-bound model was as follows [47]:

$$\log L = \sum_{i=1}^{n} \left\{ I_i I_i^u \log\left[1 - F\left((t_i^u - x_i' \ \beta)/v\right)\right] + I_i(1 - I_i^u) \log\left[F\left((t_i^u - x_i' \ \beta)/v\right) - F\left((t_i - x_i' \beta)/v\right)\right] + I_i(1 - I_i) \log\left[F\left((t_i - x_i' \ \beta)/v\right) - F\left((t_i^l - x_i' \ \beta)/v\right)\right] + (1 - I_i)\left(1 - I_i^l\right)\log\left[F\left((t_i^l - x_i' \ \beta)/v\right)\right] \right\} \tag{2}$$

where "$t_i$" is the value in the first question offered; "$t_i^u$" is the follow-up when the answer to the first question is "yes," in other words, willingness to pay the value offered; and "$t_i^l$" is the follow-up if the first answer is "no" or not willing to pay the value offered. $I_i$, $I_i^u$, and $I_i^l$ are dichotomous variables with a value of "1" if the first offer or follow-up action is positive with the answer of "yes" or dichotomous variables with a value of "0" if the response to the first question or follow-up action is negative with the answer of "no".

## 3. Results

The survey captured respondents from diverse backgrounds, focusing on primary/secondary school ages to foster environmental awareness. Demographic data Table 1 included gender, age, education, profession, and income. The average gender value was 0.493, indicating that 49.3% of respondents were male. The median values for categories of variables age, education, and professional level were 5, 3, and 2, respectively, suggesting most respondents were around 36–40 years old, had completed undergraduate studies, and were employed. The median income level was 1, indicating most earned less than IDR three million monthly. These demographics suggest individuals most likely to contribute to conservation efforts are those aged 36–40, with a bachelor's degree, employed, and earning less than IDR three million per month. This data can inform strategies and pricing schemes for conservation efforts.

**Table 1.** Socioeconomic characteristics of respondents.

| Variable | Categories | Percentages of Demographic | | Mean | Median | Quartiles | | | | |
|---|---|---|---|---|---|---|---|---|---|---|
| | | | | | | Min | Q1 | Q2 | Q3 | Max |
| gender | 0 | male | 50.35% | 0.493 | 0 | | | | | |
| | 1 | female | 48.96% | | | | | | | |
| age | 1 | 16–20 years | 14.15% | 4.491 | 5 | 1 | 2 | 5 | 6.25 | 8 |
| | 2 | 21–25 years | 13.46% | | | | | | | |
| | 3 | 26–30 years | 9.51% | | | | | | | |
| | 4 | 31–35 years | 11.37% | | | | | | | |
| | 5 | 36–40 years | 11.37% | | | | | | | |
| | 6 | 41–45 years | 14.62% | | | | | | | |
| | 7 | 46–50 years | 12.30% | | | | | | | |
| | 8 | >50 years | 12.53% | | | | | | | |
| education | 1 | primary/secondary | 19.26% | 2.521 | 3 | 1 | 2 | 3 | 3 | 5 |
| | 2 | postsecondary | 23.90% | | | | | | | |
| | 3 | undergraduate | 43.39% | | | | | | | |
| | 4 | graduate | 10.67% | | | | | | | |
| | 5 | postgraduate | 2.09% | | | | | | | |
| profession | 1 | student | 17.17% | 2.297 | 2 | 1 | 2 | 2 | 3 | 5 |
| | 2 | employed | 51.51% | | | | | | | |
| | 3 | self-emp. | 18.79% | | | | | | | |
| | 4 | unemployed | 7.66% | | | | | | | |
| | 5 | other | 4.18% | | | | | | | |

**Table 1.** *Cont.*

| Variable | Categories | Percentages of Demographic | | Mean | Median | Quartiles | | | | |
|---|---|---|---|---|---|---|---|---|---|---|
| | | | | | | Min | Q1 | Q2 | Q3 | Max |
| income | 1 | ≤2.9 M | 58.70% | 1.633 | 1 | 1 | 1 | 1 | 2 | 6 |
| | 2 | 3–5.9 M | 28.31% | | | | | | | |
| | 3 | 6–8.9 M | 7.19% | | | | | | | |
| | 4 | 9–11.9 M | 1.62% | | | | | | | |
| | 5 | 12–14.9 M | 2.09% | | | | | | | |
| | 6 | ≥15 M | 1.39% | | | | | | | |

Table 2 presents demographic correlations indicating the community's willingness to support the donation of compensation funds for forest protection and the conservation of protected wild animals. Age and profession had weak correlations, while education level and income had moderate correlations. A respondent's condition also had a negative correlation, indicating a weak correlation with education. Education had a weak correlation with environmental issues, with forest protection and protected wild animal preservation handled well by the government. Respondents had consumed protected wild animals in their lifetimes but were willing to pay or make donations for protection purposes.

**Table 2.** Correlation table of the standard and guiding statements of study.

| Variable | Gender | Age | Education | Income | S2 | S4 | S5 | S8 | S9 |
|---|---|---|---|---|---|---|---|---|---|
| age | 0.198 *** | — | | | | | | | |
| education | - | - | — | | | | | | |
| profession | −0.098 * | 0.354 *** | −0.215 *** | | | | | | |
| income | −0.115 * | 0.373 *** | 0.472 *** | — | | | | | |
| S2 | - | −0.103 * | −0.101 * | −0.108 * | — | | | | |
| S4 | - | −0.1 * | 0.158 ** | - | - | — | | | |
| S5 | - | - | - | - | 0.525 *** | - | — | | |
| S7 | 0.106 * | −0.097 * | 0.121 * | - | - | - | - | | |
| S8 | - | - | - | −0.112 * | 0.237 *** | - | 0.259 *** | — | |
| S9 | - | - | - | −0.106 * | 0.1 * | −0.198 *** | - | 0.168 *** | — |
| S10 | - | - | 0.168 *** | 0.184 *** | −0.155 ** | 0.105 * | - | −0.141 ** | −0.474 *** |

$* p < 0.05$, $** p < 0.01$, and $*** p < 0.001$. Legend: S2. I am aware of issues in the natural environment. S4. I agree that issues of the natural environment and protected animals in North Sulawesi are handled properly. S5. I am aware of protected wild animal species. S7. I have consumed one of these protected wild species during my lifetime. S8. I agree that there should be compensation. S9. I can donate funds to preserve protected wild animals with the ability to pay annually for five years. S10. I am willing to pay a certain amount per year for five years. Signs "—" and "-" does not have a different meaning.

The participants' willingness to pay for forest protection and conservation of protected wild animals was positively influenced by their income level, while their knowledge of environmental conditions and protected wildlife had a moderate correlation. They expressed agreement with the concept of a protection compensation fund and were also supportive of the idea of a compensation fund and showed a willingness to pay.

The aggregation of the survey results is shown in Table 3. Of the 428 respondents, 10 respondents did not care about environmental issues (2.3%), and three did not care about protected wild animals (0.7%). Based on the results obtained, respondents thought that the important issues faced were 70.6% environmental issues and 52.3% health issues. Natural environment issues consisted of floods at 65.1% and forest destruction at 55.5%. Meanwhile, the reduction or loss of protected wild animals was thought to be caused by hunting (85.1%), forest destruction (62.9%), and tree felling (56.4%).

**Table 3.** The aggregates were obtained from the survey results.

| | | |
|---|---|---|
| 1. Essential issues faced for the last 3 years | Environment | 70.6% |
| | Health | 52.3% |
| 3. Essential natural environmental issues faced in the past 3 years. | Flood | 65.1% |
| | Forest destruction | 55.5% |
| 6. Main cause of the decline or loss of protected wild animals. | Hunting | 85.1% |
| | Forest destruction | 62.9% |
| | Tree felling | 56.4% |

| | Strongly agree | Agree | Neutral | Disagree | Strongly disagree |
|---|---|---|---|---|---|
| 2. I am aware of issues in the natural environment. | 52.8% | 39.3% | 5.6% | 2.3% | 0.0% |
| 4. I agree that issues of the natural environment are handled properly. | 18.4% | 25.1% | 27.8% | 26.6% | 2.2% |
| 5. I am aware of the issues of protected wild animal species. | 54.1% | 41.9% | 3.3% | 0.7% | 0.0% |
| 8. I agree that there should be compensation | 53.7% | 36.1% | 8.0% | 2.2% | 0.0% |

| | Often | Ever | Hesitate | Never | Forgot |
|---|---|---|---|---|---|
| 7. I have consumed one of these protected wild animals during my lifetime. | 0.2% | 40.2% | 8.9% | 47.5% | 3.1% |

9. I can pay every year for 5 years to donate to protect forests and protected wild animal species. (Y/N) **

10. I am willing to pay a certain amount.

| bidi * | bidh * | bidl * | yy | yn | ny | nn |
|---|---|---|---|---|---|---|
| 50 | 25 | 0 | - | - | - | 31.80% |
| 50 | 100 | 25 | 26.30% | 1.00% | - | - |
| 100 | 150 | 50 | 6.00% | 4.30% | - | - |
| 150 | 200 | 100 | 5.50% | - | - | - |
| 200 | 250 | 150 | 2.40% | 4.60% | - | - |
| 250 | 300 | 200 | 1.20% | 2.90% | - | - |
| 300 | 400 | 250 | - | 1.20% | - | - |
| 400 | 300 | 250 | - | - | 1.70% | - |
| | 500 | 300 | 0.20% | - | - | - |
| 500 | 400 | 300 | - | - | 0.70% | - |
| | 750 | 400 | - | 0.70% | - | - |
| 750 | 500 | 400 | - | - | 5.50% | - |
| | 1000 | 500 | - | - | - | - |
| 1000 | 750 | 500 | - | - | 1.70% | - |
| | 1250 | 750 | - | - | - | - |
| 1250 | 1000 | 750 | - | - | 2.30% | - |
| | 1500 | 1000 | - | - | - | - |

* Bid amounts:—bidi = first bid amount offered in Indonesian Rupiah (IDR) per year.—bidh = second bid amount offered in IDR per year if bidi is rejected.—bidl = third bid amount offered in IDR immediately after bidi and bidh are rejected. Response options:—yy = "yes yes", agrees to both bidi and bidh amounts.—yn = "yes no", agrees to first bid amount (bidi) but rejects second bid amount (bidh).—nn = "no no", rejects both initial bid amounts (bidi and bidh). ** If choose "Y" then go to question 10, if choose "N" go to socio-economic questions.

For the positive results obtained for willingness to pay, 68.2% of these respondents indicated a relatively large number and high enthusiasm for preserving forests in North Sulawesi. Respondents who responded otherwise were divided into two groups: first, they did not care about the environment and protected wild animals, and second, they did not want to contribute. Of those who did not want to contribute, 31.8% thought that preserving

the forest was 1. the responsibility of the government or 2. said they would be responsible if they had a job.

Here, we show two types of statistical data, namely the WTP mean and median results from the complete summary output (Table 4). We can see that the empirical model estimation of the DBDC resulted from a 95% confidential interval for WTP estimation.

**Table 4.** WTP estimates for the Krinsky–Robb and Bootstrap confidence intervals.

| Krinsky–Robb | Estimate | LB | UB |
|---|---|---|---|
| Mean | (61.398) | (65.553) | 127,800 |
| Truncated Mean | 449,740 | 83,936 | 1,180,200 |
| Median | 264,820 | −1,083,100 | 1,460,900 |
| **Bootstrap** | | | |
| Mean | (61.398) | (61.77) | (61.02) |
| Truncated Mean | 449,740 | 445,460 | 685,280 |
| Median | 264,820 | 255,680 | 1,051,000 |

The Krinsky–Robb and Bootstrap methods displayed the estimated mean value of population (61.398), as well as the mean values in the lower bound (LB) and upper bound (UB), which are in brackets (); these do not imply negativity, but rather denote the confidence interval of the population mean value. The results obtained for the estimation of the DBDC model using both models for the WTP truncated mean value had the same result of IDR 449,740 (USD 31.02), as well as a median WTP of IDR 264,820 (USD 18.26). What distinguished these two models was the estimation of the lower and upper limits. The Krinsky–Robb model gave a wider value estimate, while that of the Bootstrap model was narrower.

From the results of the visualization of bootstrap probability model estimates of the median WTP in Figure A1 in the Appendix B, the median WTP indicated that the higher the bid price, the lower the probability of people saying "yes". When the horizontal line highlighted the 50 percent support point in the model function plot, the desired price was IDR 264,820 (USD 18.26). This price showed the community's consideration of contributing to forest monitoring to better preserve protected wild animals.

Broadly speaking, the results that refer to the potential benefits of reality are (1). Ecological and environmental awareness: In Table 3, 70.6% of respondents provided assessments on ecological and environmental issues, including 85.1% citing problems with hunting followed by forest destruction and tree felling. The majority of respondents (92.1%) stated that they are aware of environmental issues, including 96% who are aware of the protection of wild animals. This indicates a high level of awareness about these issues among the local community. (2). Socioeconomically, as many as 89.8% of respondents agreed or strongly agreed with the idea of a compensation fund to protect forests and wild animals. This shows strong support for this approach. (3). Legal and policy aspect, respondents opined that ecological issues have not been or are not being handled well by the government, with 43.5% holding this view compared to 28.8% who felt otherwise. This can also be seen from the still high percentage of respondents who ever consumed wild animals, at 40.2%, with 0.2% still doing so regularly. (4). Monetary Value: The monetary value determined by respondents for forest services is IDR 264,820 (USD 18.26) per year. This provides an estimate of how much the local community is willing to pay to protect forests and wild animals.

## 4. Discussion

This study addressed the rapid degradation of forests in North Sulawesi from human activities, which threatens wildlife and ecological stability. Our findings align with previous research highlighting the multifaceted dimensions of this issue, including ecological, socioeconomic, and policy aspects [48]. In terms of ecology, we confirmed the vital role of

forests in climate regulation and maintaining biodiversity via complex ecosystem services. Socioeconomically, our study reinforced that local communities depend on forests but also drive deforestation and poaching for economic reasons, consistent with past research [49]. Regarding policy, we found that despite the existence of environmental protection for forests and wildlife, implementation and enforcement remain challenging, echoing the conclusions of other studies [50].

A key knowledge gap we addressed was the lack of research on the social-monetary valuation of conservation initiatives in North Sulawesi specifically. Our study pioneered the assessment of monetary values local communities are willing to contribute towards conservation. These novel findings can inform policymaking by providing evidence to develop ecologically sound, socially accepted, and economically viable conservation strategies supported by strong legal policy aspects. This ultimately enhances the likelihood of successful, sustainable conservation outcomes in North Sulawesi, consistent with principles of conservation policy design [50]. Further research can build on these findings by expanding the valuation to other stakeholders and assessing the impact of conservation funding on deforestation rates and ecological indicators over time.

This study was conducted out of concern for the destruction of natural environments where protected wild species are located. We conducted a study to assess how much attention a community pays to protected wildlife ecosystems threatened with extinction in their degraded environment. This study found that there were 68.2% of respondents made the decision to be able to donate, and the rest chose non-willingness to pay. We think this is a reasonably large number, and some of the motivations they expressed in the survey input column were that environmental sustainability was the government's responsibility [43,51] or that they would contribute if they had a job. These statements have similarities with the study of Opacak and Wang (2019), where respondents said that environmental issues were the government's responsibility, that they could not afford the additional burden, and that they did not believe their money would be used by the agreement [39].

This paper was a study on the monetary valuation of protected wild animals. We divided the study into two parts by ensuring that the statement/question questionnaire and the data, methods, and materials were different. This difference was because we had two different output objectives: one was to determine the mean price of protected wild animals, and the other was to determine the median price of willingness to pay for forest protection and protected wild animals. The same data were only the demographic data and some referral questions. This is worth performing, as Fine and Kurdek (1994) revealed in their paper about publishing multiple journal articles from a single data set [52].

After we ran the extended DBDC model on the factors as a proxy for familiarity with the amount of willingness to pay set at the median estimate of IDR 264,820 (USD 18.26), this WTP amount could be contributed annually for five years. Furthermore, the correlation analysis results showed that demographic estimates had a moderate correlation between education and income factors. This result is in line with a study conducted by Adrian and Khoirunurrofik (2021) [53] but not with that of Wolla and Sullivan (2017), who found this correlation to be very strong [54]. Meanwhile, a low positive association correlation occurred in employees aged 36–40 years with an average income of less than IDR three million per month. The correlations between education and income, age and profession, and age and income were better than the correlations between other factors. We also found a strong correlation among respondents who were aware that if there was a problem in the environment, there was a problem with protected wild species. It did not escape our observation that there was a moderately strong negative correlation ($-0.474$ ***) between statement 9 (S9) regarding yes/no for the decision of willingness to pay and statement 10 (S10) regarding the decision of how much to give. We have to explore this result in a follow-up study. If the number of respondents increases, there may also be an increase in those who do not want to donate, and the estimated median WTP price may increase for those who want to donate, or vice versa.

Some inputs from the survey results that can be summarized in the discussion are that respondents were grateful for understanding the state of the natural environment. They also encouraged the maintenance and improved supervision of forests so that their sustainability was maintained and protected wild animals did not become extinct. They asked that counseling be conducted more often, even at the student level. Regarding the willingness to pay, several respondents felt that this was the government's responsibility. They agreed that there should be a compensation fund for forest damage, but that did not mean that this burden was the community's responsibility.

An obstacle faced by paper-based surveys for research during the COVID-19 pandemic was that access to cities, and regencies was complicated because of various procedures that had to be followed. Meanwhile, online obstacles were that some places did not have qualified internet access. In addition, some respondents were not proficient enough in using existing media, and some devices used did not support access to the questionnaire. The main limitation we found in this study was that we did not determine the payment method, such as assigning or entrusting existing institutions as managers and depositories of funds or forming new institutions. This limitation can be a recommendation for further research. Future significance could include direct community involvement to protect forests, for social control, and for assisting governments in preserving the natural environment because environmental problems are not only the responsibility of the government alone but are the responsibility of all [55].

We may be able to identify several limitations, such as the sample size, which only uses native people or those born and raised in North Sulawesi. Additionally, the following limitation is that the research was conducted for approximately 6 months during the COVID-19 pandemic, which may not represent the long-term impact of the intervention.

Despite these limitations, the study will provide valuable insights into the willingness of people in North Sulawesi to pay for forest protection. It will also provide information on the monetary value that people place on forest protection, which can inform policy decisions on forest conservation in North Sulawesi.

Strategic interventions are needed to address the complex drivers of deforestation and wildlife threats in North Sulawesi. Our findings suggest four key approaches. First, increase education and awareness about sustainable development and environmental impacts. Second, encourage community participation in resource management decisions. Third, develop policies and regulations that support sustainability, such as prohibiting illegal logging. Finally, utilize technology to increase efficiency and reduce human footprint. Further research can elucidate social, economic, and environmental interactions influencing forest degradation, providing insights to refine solutions. If strategic interventions are not implemented, we risk facing a continuation of the status quo, as evidenced by the ongoing trends of degradation and deforestation. Proactive measures must be taken to change the current trajectory and safeguard these vital forest ecosystems.

This study offers crucial insights into North Sulawesi's local community's perception of environmental and wildlife conservation, highlighting key concerns such as flooding, forest destruction, and hunting-induced wildlife decline. The findings, revealing high environmental awareness and a willingness to contribute to conservation financially, can guide the creation of effective conservation strategies that have community support. These strategies could include awareness campaigns, hunting regulation, sustainable land use promotion, and conservation funding mechanisms. The research could assist government bodies, NGOs, and local communities in formulating and implementing robust conservation strategies by identifying areas most susceptible to deforestation or species most threatened by poaching. It also provides the scientific evidence necessary to support new policies or regulations aimed at forest and wildlife protection.

## 5. Conclusions

The results of the contingent valuation in this study showed that a high level of community participation contributed to environmental and forest sustainability. Concern

for integrated supervision needs to be addressed. This can be proved by the portion of respondents who agreed and even strongly agreed that there should be compensation for forest environmental monitoring, reaching 89.8%. From there, 67.5% said they were able to pay, although 31.8% said they were unable to pay for the reasons that it was the responsibility of the government and that they would be responsible if they had a job. For the effectiveness of obtaining compensation funds, two things were found: 1. a need for the socialization of current conditions to make people aware of the urgency of the environment, and 2. targeting specifications for the education and income factors.

The study found that the median willingness-to-pay (WTP) of respondents willing to spend money to conserve forests and protect wildlife ranged from IDR 255,680 (USD 17.63) to IDR 1,051,000 (USD 72.48), with a 50% probability support point at IDR 264,820 (USD 18.26).

The study found that a sizable portion of the community surveyed expressed willingness to pay for forest protection and conservation of protected wild animals. Respondents believed that environmental issues were important, with floods and forest destruction being the most pressing natural environmental issues. The study highlights the importance of tailored pricing schemes and strategy approaches based on the demographic characteristics of the target population. The findings suggest that designing compensation funds for forest protection and conservation of protected wild animals could be effective in encouraging community involvement in preserving the natural environment by emphasizing the importance of community involvement in protecting forests and preserving the natural environment. The findings suggest that designing compensation funds for forest protection and conservation of protected wild animals could be an effective way to encourage community participation. However, further research is needed to address the limitations of the study and develop more effective strategies for promoting environmental preservation.

**Author Contributions:** Conceptualization, J.M. and Y.H.; Methodology, J.M. and Y.H.; Validation, J.M. and J.H.; Formal Analysis, J.M.; Investigation, Y.H. and J.H.; Resources, J.M.; Data Curation, Y.H.; Writing—Original Draft Preparation, J.M.; Writing—Review and Editing, J.M., Y.H., and J.H.; Supervision, Y.H. and J.H.; Project Administration, J.M., Y.H. and J.H.; Visualization, J.M. and J.H.; Funding Acquisition, Y.H. All authors have read and agreed to the published version of the manuscript. All authors contributed to the paper.

**Funding:** This research received no external funding.

**Data Availability Statement:** The data presented in this study are available on request from the corresponding author. The data are not publicly available due to privacy.

**Acknowledgments:** 1. The author would like to thank his colleagues for their assistance in this study. Despite the COVID-19 pandemic, the following helped distribute the questionnaire online and offline: Berny Antouw, Novi Katiandagho, Billy Sakul, Yusuf Kalengkongan, Jerry Simbar, Fabyo Rumagit, Andrew Antouw, Edner Deeng, Ronald Mangamis, Hartje Sandil, Stanley Moningka, and Agus Sahensolar. Map image: Muhammad Rifky Rayna. Additionally, the author would like to convey gratitude to his statistician mentor, Amelia Tanasale. 2. Indonesian Resources and Development Institute (IRDI).

**Conflicts of Interest:** The authors declare no conflict of interest.

## Appendix A. Additional Tables

**Table A1.** Respondent data by region.

| Region | Respondents |
| --- | --- |
| 1. Bitung Municipality | 32 |
| 2. Kotamobagu Municipality | 0 |
| 3. Manado Municipality | 51 |
| 4. Tomohon Municipality | 58 |
| 5. South Bolaang Mongondow Regency | 30 |
| 6. East Bolaang Mongondow Regency | 1 * |
| 7. North Bolaang Mongondow Regency | 0 |
| 8. Sangihe Regency | 20 |
| 9. Talaud Regency | 60 |
| 10. Minahasa Regency | 52 |
| 11. South Minahasa Regency | 24 |
| 12. Southeast Minahasa Regency | 59 |
| 13. North Minahasa Regency | 25 |
| 14. Siau Tagulandang Biaro Regency | 0 |
| 15. Bolaang Mongondow Regency | 16 |
| Total | 428 |

* One respondent from East Bolaang Mongondow Regency participated through an online questionnaire.

**Table A2.** Structure of questionnaire statements (S).

| Standard Statement Questionnaire | Guiding Statement to WTP | WTP Main Statement (DBDC) |
| --- | --- | --- |
| 1. What is an essential issue faced by North Sulawesi for the last 3 years? | 6. What is the main cause of the decline or loss of protected wild animals in North Sulawesi? | 10. By answering "yes", I am willing to pay a certain amount per year for five years that I can donate to protect forests and protected wild animals. |
| 2. I am aware of issues in the natural environment. | 7. I have consumed one of these protected wild species during my lifetime. | |
| 3. What is an essential natural environmental issue that North Sulawesi has faced in the past 3 years? | | |
| 4. I agree that issues of the natural environment and protected animals in North Sulawesi are handled properly. | 8. I agree that there should be compensation that I/the government/other parties should give to people living around the forest as a form of responsibility to maintain the balance of forests by not hunting protected wild animals. | |
| 5. I am aware of the issues of protected wild animal species. | 9. If the habitat situation becomes worse and endangered animals are headed for extinction, I can donate funds to preserve protected wild animals with the ability to pay annually for 5 years. | |

## Appendix B. Additional Figures

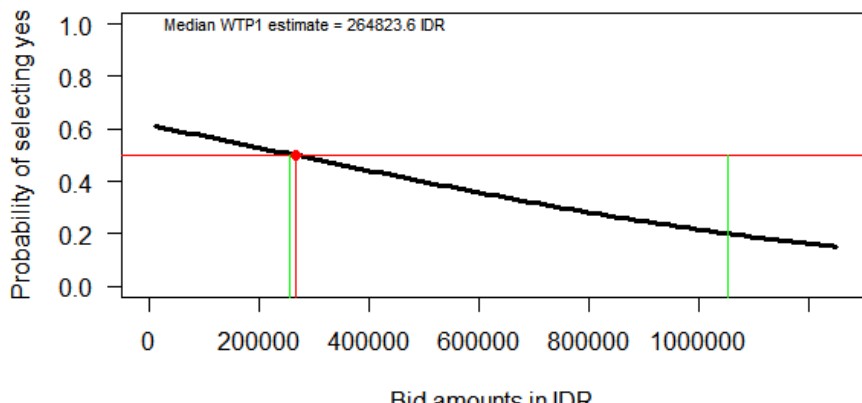

**Figure A1.** Bootstrap probability model estimate of the WTP median. The horizontal red line is the 50% midian probability, the vertical red line is the midian estimate, the green line shows the median lower bound (LB) and upper bound (UB). The black line is the monetary rupiah value the respondent is willing to pay.

## Appendix C. Additional Method

Stage 1. Preparation of the questionnaire.

The willingness-to-pay questionnaire was designed in two sections: namely, the characteristics of the forest area to be protected in relation to the preservation of protected wild animal species and the socioeconomic characteristics of the respondents. The requirements to become a respondent were as follows: 1. a North Sulawesi native living in North Sulawesi, 2. an individual of non-North Sulawesi ethnicity who was born, raised, and living in North Sulawesi, or 3. an individual of non-North Sulawesi ethnicity who had lived in North Sulawesi. The general criteria for respondents included the following: over 16 years of age and native to North Sulawesi (born and raised) or having lived in North Sulawesi for 15 years. In the forests of North Sulawesi live at least nineteen families of protected wild animals, consisting of eight families of mammals, nine families of birds, and two families of reptiles [8].

The research tool was a questionnaire survey [33]. The statements of the survey questions were as shown in Table A2 in the Appendix and included scenario statements so that a respondent was directed to what was occurring in the environment in which he was involved. The survey consisted of 10 questions assessing respondents' perspectives on various conservation issues in North Sulawesi. For questions 1, 3, and 6, participants were asked to select their top 3 choices from a list of options. We analyzed only response choices exceeding >50% of total responses to highlight the most selected issues. Questions 2, 4, 5, and 8 utilized 5-point Likert scale responses ranging from "strongly disagree" to "strongly agree". We combined "agree" and "strongly agree" responses for analysis and reporting of these questions. Question 7 asked respondents to select their typical actions regarding protected wildlife species, while question 9 served as a gateway to determine who would answer the follow-up question 10 regarding willingness-to-pay bid amounts for conservation initiatives. This structured approach allowed quantitative analysis of key perspectives on conservation priorities, behaviors, and economic valuations in the study population. The first part consisted of standard statements that directed respondents to give their current opinions about their views on the actual condition of the ecosystem. Questions were designed so that respondents could give their best opinion before they were asked to contribute [56]. The second part was a decision question, and the third part was the choice of an amount that they were ready to give over five years. There was also a statement of the extent to which respondents understood the current situation so that

they could make decisions. The socioeconomic characteristics of the participants were assessed based on their age, gender, education, profession, and income. Age was divided into two categories, male and female, and into eight age groups. Education was categorized into five levels of attainment. Professions were classified into five categories. Income was categorized into six groups based on the income level.

Stage 2. The process of distributing the sample questionnaire.

Questionnaires were distributed on a paper basis and online. In this study, a purposive sampling method [57] was selected to determine participation in both paper and online surveys. A total of 400 paper questionnaires were distributed, of which 364 were returned. Out of these, 342 questionnaires met the required standards and were included in the analysis. A modified Dillman approach was used for the online survey [58], which yielded 86 responses. Offline and online questionnaires have the same questions but differ in distribution. We considered online surveys as a response to the challenges of access to respondent locations posed by the COVID-19 pandemic. The data collection process was carried out for three months, with distribution techniques discussed within the team (acknowledgment). Then, the questionnaires were processed using the contingent valuation method to estimate the values of goods or services with no market price. Practically, this method raises an interesting question: if a habitat situation worsens and endangered animals are headed for extinction, are individuals willing to pay every year to keep the animals from becoming extinct? The study was structured as a willingness-to-pay evaluation using the rupiah (IDR), and the USD-to-IDR exchange rate was 14,500 (April 2021). The population of respondents was spread from urban areas to remote villages, and its membership consisted of various economic statuses, occupations, education levels, and social strata. In this way, we measured a sample that was considered representative, as determined using the Slovin formula sampling method [59].

Stage 3. Data analysis and processing.

Then, the data were processed using a descriptive statistical method. This method interpreted demographic characteristics and correlations between the variables. Next, we identified the real possibility of correlated behavior parameters. The procedure was implemented in the JASP v.0.13.01 script [60]. The study only described variables that were significantly correlated and ignored others, making an argument for formulating the conclusions of the study subject. The contingency valuation method (CVM) [61] was also used based on the well-known consumer economic theory, where individual values reflect individual preferences—or pleasure or well-being—according to the constraints perceived by consumers. Using a purposive sample of surveys from the relevant population, the CVM aimed to create a hypothetical market that allowed respondents to express the use of values that could not exist at previous market values. The contingency valuation approach was helpful in estimating the economic worth of the environment and natural resources despite its flaws, but it needed to be applied with caution. [62,63].

The CVM represents value in monetary terms and is always associated with data [64]. This model is used in business-to-business market situations where products, services, and offerings are based on the amounts that customers create for value. Thus, customer benefits can be measured in the CVM via features and products in the form of money; the specificity of this model seeks to measure the values of suppliers and customers. This method has been widely applied using the basic concept in contingency assessment, where the value of consumer information is widely used in assessing nonmarket public goods against private valuations. In combination with the empirical assessment process in the open survey, the dichotomous choice contingency valuation method DCCVM was adopted [36]. Furthermore, interpretation was performed using statistical descriptions. This method focused on the practical skills required to study contingent valuation using the standard method of a (DBDC) format [40].

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
