# Peer review of "Social Monetary Valuation for Protecting Forests and Protected Wild Animals in North Sulawesi, Indonesia"

_forests, doi:10.3390/f14102114_

Round 1
Reviewer 1 Report (Previous Reviewer 2)
No suggestions. Good work.
Author Response
Please see the attachment

Reviewer 2 Report (New Reviewer)
The authors need to attend the following suggestions in order to support the the relationship among introduction, objectives, methods, results, discussion, and conclusions:
They need to improve the introduction, literature review, and objectives. They must include more international and national background about the research topic. Specify with more details the study problem and justification (multidimensional).
Precise methodology: How did they define sample size?. They need to support it.
The results must be explanatory in reference to the reality of the potential benefits.
The discussion requires more strong support with others researchers and experiences related to the topic reserach in order to strength the discussion and conclusions, including more international and national references. They need to discuss more the desired real impacts of this kind of actions with criteria of sustainable development.
Given some limitations of the research, Can that results be generalized in any case, it is necessary to strengthen the research and its results
It is necessary to include multidimensional strategies proposals to guarantee the aplication of th obtained results in order to contribute to sustainable development.
In addition, they need to discuss what it would happen if the problem is not attended, considering social, ecological, economics, legal and environmental policy impacts.?
What are the strategies proposal in order to promote and advancing toward a real application of the results?? sustainable development as a total system?.
What is the true contribution to scientific knowledge and the solution of the real problem?
What is the practical use of the obtained results from this study? What is the true contribution to knowledge fontier and the solution of real problems?
Author Response
Please see the attachment

Reviewer 3 Report (New Reviewer)
The paper raises the interesting topic of citizens' willingness to contribute to the costs of environmental protection. An important topic during the coming period of increasing changes related to global warming and pending changes in people's model of functioning, to a more sustainable towards the environment.
Unfortunately, the work has a number of shortcomings that prevent a positive evaluation. The methods section contains very detailed descriptions of methods that could be moved to the appendixes. So does the incomprehensibly long (80 lines) description of the single statistical method used. The results section has elements of interpretation and description of statistical methods. This needs to be cleaned up
Results - the biggest shortcoming of the work, very heavily descriptive, long sections of text. In many places, the text duplicates percentages from tables and miscalculated mean values. Instead of real values of the analyzed parameters, artificial classes were used for averaging, which makes no sense (for example, the value of 2.5 ages says nothing as opposed to 39.5 years). Further - illegibly prepared tables, unclear composition and content of the tables, sparse descriptions of the contents and the introduction of abbreviations nowhere in the tables. The method of correlation of single variables used illustrates residual relationships, a better method would be one that combines all variables in a single analysis.

English needs correction.
Round 2
Reviewer 2 Report (New Reviewer)
I have reviewed the author´s responses and the manuscript has been
sufficiently improved to warrant publication in Forests.
Author Response
Please see the attachment.

Reviewer 3 Report (New Reviewer)
The manuscript has been largely revised. Fragments marked in the text still need to be corrected.

English needs to bee improved.
Author Response
Please see the attachment.

This manuscript is a resubmission of an earlier submission. The following is a list of the peer review reports and author responses from that submission.
Round 1
Reviewer 1 Report
There are three main problems
First, the english writing is of low quality which makes it difficult to understand the importance of the analysis and in some cases exactly what the analysis and valuation concerns. Words are used inappropriately such as "the scope of animals to be narrower" (line 29), which probably refers to endangered species having a reduced habitat size. I woulkd recommend that the authors use a service such as "grammarly" or pays for editorial work. In my experience, grammarly works quite well.
Second, the citations and literature used to justify the method are not appropriate. This is evident throughout but particularly around page 5 of 14. The citation [26] should not be used to justify the DCCVM method. Citations [28] and [29] should not be used to justify the discussion of biases and [30] should not be used to justify ignoring them. Citation [27] seems appropriate but there are much more fudnamental citations for biases and emthods in contingent valuation and they must be cited to give the reader confidence that the correct procedure has been followed. In addition, there are citations that have been taken from other articles that are not approrpiately cited or included in the reference list. For example, lines 241-2 cite two articles taken from [27] but not in the refernce list and not cited consistently. Again, this causes the erader to be skeptical and lack confidence in the analysis in the paper.
Third, with regard to the analysis itself, it seems that the correct procedures have been followed to an extent. However, the whole setup of the experiment to value forest protection does not appear to be appropriate. This may be due to the english language but as a reader I stuggle to see what people are being asked to value. There does not seem to be a consistent action or good or service that the respondents are vbeing asked to value. The respondents seem to be asked about their experience with the forests and asked to value something vaguely like "protection" and in another case "monitoring" (line 340). As a reader ask: protection of what? The forest? That small part of the forst that the respondent has experienced? The resources from the forest? The endagered species? The food? What is being valued here?
This speaks to the method of contingent valuation and the biases. Without a clear picture of what is being valued, the contingent valuation method is not able to accurately value anything. The major issue here is the embedding bias. People will be responding to something vague and unknowable and will generally be valuing something very broad. In any case, each respondent will be valuing something different and this is not a valid approach for contingent valuation. Reading the main literature in contingent valuation, rather than other studies that use contingent valuation (citations mentioned above), will help the authors understand this point.
Writing and referencing can be fixed. However, in my opinion, the setup of the contingent valuation experiment is innapropriate and can't be fixed at this stage. The authors would neeed to start again and set up a consistent good or service or conservation action to be valued.
As above
Reviewer 2 Report
Dear Authors!
This paper is interesting but with the consideration of the following concerns:
1. Can you cite more papers using the econometric model? mathematical formula: (1), (2).?
2. Table 1. For qualitative variables, it is better to present numbers and percentages. In the table, the mean and SD should refer to the actual values ​​of descriptive statistics for quantitative variables. It may be better to use the median and quartile range, eg for the variable "age"?On the other hand, providing the mean and SD for the age and income variables raises objections. For example, is the mean of Age=4,491, and Income=1,633?
3. In the line 272 You wrote: “The mean age is between 31 to 40 years,”. Maybe it's better to present the specific value of the variable age and others in the text of the article?
4. In the line: 241-243: „The econometric model we adopt is proposed by Cameron and James (1987) and 241 Cameron (1988), with this model allowing us to calculate the mean or median value of WTP easily [27]”. Please check the source and year of publication of the article.
Please describe the specific references of the applications of the selected methods to solve similar problems. Please consider supplementing with application literature about econometric methods?
5. In the line 341-395: Only three bibliographic items [30], [31], [32] were cited in the discussion. Maybe the Authors will write more about this.
6. In the line: 153-155 . Is the map in Figure 1 created by you? If not, you must cite the source and have the consent of the Authors (Publisher).
Please indicate the correctness of the selection of the research sample (random, proportional or other?). Please check the calculations and results. The work should be carefully edited. Some references are missing page numbers. Consider citing more references, especially on methods and discussions.
Reviewer 3 Report
The authors present a contingent valuation study set in North Sulawesi. In my opinion, contingent valuation studies in highly biodiverse region are potentially valuable.
The authors can do a better job of placing their paper in the literature. Currently, the introduction is underreferenced. Consider including Diaz et al. (2019), Santika et al. (2022), or Sol (2019). The authors can also provide a more extensive discussion of the rich contingent valuation literature (e.g., Diamond and Hausman, 1994; Kling et al., 2012) and become more specific on how their study contributes to the literature.
While much of the study design seems sound, the presentation of various design elements and descriptives should be improved before the study merits publication. For example, the authors describe that surveys have been distributed on paper and online; how were the surveys distributed? how many paper surveys were sent? what share of the responses came from paper surveys? were the online surveys sensitive for completion by bots?
Table 1 could include survey measures. The discussion of Table 2 could be shortened, while discussion of Table 4 should be more extensive; e.g., what do parentheses mean? A footnote to the table could be useful.
References
Diamond, P.A. and Hausman, J.A., 1994. Contingent valuation: is some number better than no number?. Journal of economic perspectives, 8(4), pp.45-64.
Díaz, S., Settele, J., Brondízio, E.S., Ngo, H.T., Agard, J., Arneth, A., Balvanera, P., Brauman, K.A., Butchart, S.H., Chan, K.M. and Garibaldi, L.A., 2019. Pervasive human-driven decline of life on Earth points to the need for transformative change. Science, 366(6471), p.eaax3100.
Kling, C.L., Phaneuf, D.J. and Zhao, J., 2012. From Exxon to BP: Has some number become better than no number?. Journal of Economic Perspectives, 26(4), pp.3-26.
Santika, T., Sherman, J., Voigt, M., Ancrenaz, M., Wich, S.A., Wilson, K.A., Possingham, H., Massingham, E., Seaman, D.J., Ashbury, A.M. and Azvi, T.S., 2022. Effectiveness of 20 years of conservation investments in protecting orangutans. Current Biology, 32(8), pp.1754-1763.
Sol, J., 2019. Economics in the anthropocene: species extinction or steady state economics. Ecological Economics, 165, p.106392.
The paper needs extensive English editing, too much to sum up here. The level of English was also concerning with respect to the survey items; I doubt that "I agree that there was compensation" would accurate translate back to the original survey item.
Line 156-165 is double.
Round 2
Reviewer 3 Report
While suggested references were integrated, their description does little service to the potential readers.
Try to become more explicit to avoid vague or unclear statements; e.g., "Their income factor supported this willingness"
